# Molecular and Dynamic Evaluation of Proteins Related to Resistance to Neoadjuvant Treatment with Chemoradiotherapy in Circulating Tumor Cells of Patients with Locally Advanced Rectal Cancer

**DOI:** 10.3390/cells10061539

**Published:** 2021-06-18

**Authors:** Virgílio Souza e Silva, Emne Ali Abdallah, Bianca de Cássia Troncarelli Flores, Alexcia Camila Braun, Daniela de Jesus Ferreira Costa, Anna Paula Carreta Ruano, Vanessa Alves Gasparini, Maria Letícia Gobo Silva, Gustavo Gomes Mendes, Laura Carolina Lopez Claro, Vinicius Fernando Calsavara, Samuel Aguiar Junior, Celso Abdon Lopes de Mello, Ludmilla Thomé Domingos Chinen

**Affiliations:** 1Department of Medical Oncology, A.C.Camargo Cancer Center, São Paulo 01509-900, Brazil; virgilio.silva@accamargo.org.br (V.S.e.S.); celso.mello@accamargo.org.br (C.A.L.d.M.); 2International Research Center, A.C.Camargo Cancer Center, São Paulo 01508-010, Brazil; emne.abdallah@accamargo.org.br (E.A.A.); biancatroncarelli8@gmail.com (B.d.C.T.F.); alexiabraun@gmail.com (A.C.B.); danielajfcosta07@gmail.com (D.d.J.F.C.); annacarreta@gmail.com (A.P.C.R.); vanessa.alves@accamargo.org.br (V.A.G.); viniciusfcalsavara@gmail.com (V.F.C.); 3Department of Radiotherapy, A.C.Camargo Cancer Center, São Paulo 01509-900, Brazil; leticia.silva@accamargo.org.br; 4Department of Radiology, A.C.Camargo Cancer Center, São Paulo 01509-900, Brazil; gustavo.mendes@accamargo.org.br; 5Department of Pathology, A.C.Camargo Cancer Center, São Paulo 01509-900, Brazil; laura.claro@accamargo.org.br; 6Department of Pelvic Surgery, A.C.Camargo Cancer Center, São Paulo 01509-900, Brazil; samuel.aguiar@accamargo.org.br; 7National Institute for Science and Technology in Oncogenomics and Therapeutic Innovation, São Paulo 01509-900, Brazil

**Keywords:** locally advanced rectal carcinoma, neoadjuvant chemoradiation, liquid biopsy, circulating tumor cell, RAD23B

## Abstract

The heterogeneity of response to neoadjuvant chemoradiotherapy (NCRT) is still a challenge in locally advanced rectal cancer (LARC). The evaluation of thymidylate synthase (TYMS) and RAD23 homolog B (RAD23B) expression in circulating tumor cells (CTCs) provides complementary clinical information. CTCs were prospectively evaluated in 166 blood samples (63 patients) with LARC undergoing NCRT. The primary objective was to verify if the absence of RAD23B/TYMS in CTCs would correlate with pathological complete response (pCR). Secondary objectives were to correlate CTC kinetics before (C1)/after NCRT (C2), in addition to the expression of transforming growth factor-β receptor I (TGF-βRI) with survival rates. CTCs were isolated by ISET and evaluated by immunocytochemistry (protein expression). At C1, RAD23B was detected in 54.1% of patients with no pCR and its absence in 91.7% of patients with pCR (*p* = 0.014); TYMS^−^ was observed in 90% of patients with pCR and TYMS^+^ in 51.7% without pCR (*p* = 0.057). Patients with CTC2 > CTC1 had worse disease-free survival (DFS) (*p* = 0.00025) and overall survival (OS) (*p* = 0.0036) compared with those with CTC2 ≤ CTC1. TGF-βRI expression in any time correlated with worse DFS (*p* = 0.059). To conclude, RAD23B/TYMS and CTC kinetics may facilitate the personalized treatment of LARC.

## 1. Introduction

Colorectal carcinoma is the second most common cause of death in the USA and 30% of new cases are rectal carcinoma (RC) [1,2]. At diagnosis, about half of all patients with RC are diagnosed with locally advanced rectal carcinoma (LARC), which comprises, according to the TNM classification, tumors in clinical stages II and III, defined as cT3 or cT4 and/or presence of regional lymph nodes compromised by the tumor [3,4]. The standard of care for patients with cT3/T4 or N+ is neoadjuvant chemoradiation (NCRT) followed by total mesorectal excision (TME) based on the results of the German Rectal Cancer Study CAO/ARO/AIO-94 [5]. Despite significant decrease in local recurrence rates from 10% to 5%, distant metastasis and treatment morbidity are still a major problem [6,7].

More recently, two phase III trials demonstrated an improvement in disease free survival (DFS) for patients treated with total neoadjuvant therapy (TNT) when compared to conventional NCRT [8,9]. Distal rectal carcinoma is even more challenging since the great majority of patients are eventually submitted to rectal amputation and definitive colostomy, generating a high social and psychological impact. As a result, those patients treated with NCRT that present complete clinical response could be watched and have a sphincter sparing approach [10,11].

The development of accurate tools to better select patients to more aggressive therapy is paramount in rectal carcinoma. Given the need to stratify patients into responders and non-responders to NCRT prior to its onset, improving the selection of those most likely to obtain a pathological complete response (pCR), several studies have evaluated possible biomarkers, which allow the dynamic monitoring of the disease [12,13]. However, the results are still controversial with limitations for use in clinical practice.

Liquid biopsy represents a promising tool, in the prognostic evaluation and therapeutic resistance, as circulating tumor DNA (ctDNA) and/or circulating tumor cells (CTCs) can be used in a punctual and/or dynamic way during treatment, complementing the evaluation of the disease [14,15,16]. Sun et al. demonstrated in a study with 115 patients with LARC that CTCs detection is a powerful tool to assess and predict response to NCRT [17]. In a preliminary study with 30 patients, we showed that thymidylate synthase (TYMS) and RAD23 homolog B (RAD-23B) expression in CTCs before and after NCRT could predict tumor response in LARC patients [18].

Numerous previous studies, in both experimental and clinical trials, associated the overexpression of TYMS with possible resistance to fluoropyrimidine-based chemotherapy in various types of cancer [19,20]. In rectal cancer, previous studies have demonstrated that the low expression TYMS gene correlates with the pathological response to 5-Fluorouracil (5-FU)-based NCRT [21,22].

The identification of LARC patients with resistance to radiotherapy is still an obstacle. RAD23 homolog B (RAD23B) is a DNA repair protein, which is the reason why some studies correlate its expression with resistance to ionizing radiation [23,24]. RAD23B expression is elevated in tumors specimens [23,25], but in rectal cancer, so far, there are no studies that evaluate this possible marker for prediction of response.

Furthermore, understanding the tumor microenvironment formed by interactions among tumor cells, immune system, endothelial cells and cytokines is essential to determine possible therapeutic strategies and prognostic evaluation [26]. The expression of programmed cell death ligand-1 (PD-L1) in rectal cancer is still poorly studied, but some authors have shown that radiotherapy can stimulate its expression, although its prognostic/predictive impact is still unknown [27,28,29,30]. The transforming growth factor-β (TGF-β) is one of the numerous cytokines found in the tumor microenvironment and plays an important role in the tumor progression, increased angiogenesis and suppression of the immune response [31]. Thus, evaluating PD-L1 and TGF-β receptor I (TGF-βRI) expression during treatment may be significant for prognostic/prediction and better therapeutic definition.

Finally, given the first results previously presented (Troncarelli et al., 2019), we followed and expanded the sampling of these patients as an attempt to strengthen these results and structure ideas for future clinical trials that incorporate liquid biopsy in the risk stratification and classification of tumors, customizing the best therapeutic strategy for each patient with LARC.

## 2. Materials and Methods

### 2.1. Patient Population and Study Design

This is a single center, prospective, observational study carried out at A.C. Camargo Cancer Center, São Paulo, Brazil, with patients with LARC. Inclusion criteria were rectal adenocarcinoma, located up to 8 cm from dentate line by flexible proctoscopy. LARC was classified according to the TNM classification: stage cT3-T4 and/or N+, as defined by magnetic resonance imaging (MRI). The pre-treatment evaluation included a pelvic MRI and a computed tomography (CT) scan of the chest and abdomen, evaluated by a multidisciplinary team. Those patients considered candidates for NCRT by institutional protocol followed by total mesorectal resection (TME) were included. Patients submitted to different neoadjuvant regimens, as short course radiation therapy or TNT were excluded from this analysis. Other inclusion criteria were normal hematological parameters and normal renal function. Exclusion criteria were use of anticoagulation and presence of distant metastasis. 

For the study, three serial blood samples were collected. The first sample was collected at the moment of patient inclusion (baseline) before the start of NCRT (collection 1, C1). The second sample was performed after the end of NCRT and before surgery (collection 2, C2), and the third sample was performed between 2 to 4 weeks after surgery (collection 3, C3). Venous blood was collected from the antecubital vein and these samples were stored in the room temperature for a maximum of 6h before analysis (Figure 1). 

The primary outcome of this study was to determine whether the absence of RAD23B and TYMS expression in CTC would correlate with pCR to NCRT and thereby identify possible responders to treatment. The secondary outcomes were to assess the CTC kinetics between baseline (C1) and after NCRT (C2) in addition to the evaluation of the immune response markers expression such as TGF-βRI and PD-L1 with recurrence and survival rates.

### 2.2. Treatment

Patients were treated following the therapeutic recommendations of the institutional protocol. All patients received neoadjuvant radiation therapy with a conformed three-dimensional (3D) technique with 45 Gy in 25 fractions to the pelvis. In addition, a boost of 5.4 Gy was administered to the primary tumor and involved lymph nodes in three fractions, for a total of 50.4 Gy applied in 28 fractions. Chemotherapy regimens consisted of intravenous administration of 5-FU with a dose of 1000 mg/m^2^ on days 1 to 5 during weeks 1 and 5 of radiation therapy; or oral capecitabine administered at a dose of 1650 mg/m^2^/d during the entire radiation treatment period, depending on the choice of the attending physician. TME surgery were performed about 8–12 weeks after completion NCRT.

For initial assessment of all patients, the clinical examination included digital rectal examination, evaluation of routine laboratory tests such as complete blood count (CBC), liver and kidney function, carcinoembryonic antigen (CEA) level and CA 19.9, colonoscopic examination including biopsy, CT of the chest plus abdomen and pelvic MRI.

### 2.3. Ethics

This prospective study was approved by the local ethics committee (2141/15C). Written informed consent was obtained from all patients prior to study enrollment. This study was registered at ClinicalTrials.gov (NCT: 02979470).

### 2.4. Statistical Analysis

Initially, the baseline characteristics were expressed as absolute and relative frequencies for qualitative variables and as the median, minimum and maximum for quantitative variables. In order to evaluate a possible association of qualitative variables in relation to the response (Tumor Regression Grade), the independence test (Fisher’s exact test, chi-square test or chi-square test with Yates’ continuity correction) was applied to the data.

The univariable and multivariable logistic regression models were fitted to the data in order to identify possible independent factors for the occurrence of the response, in which the measure of association is given by the odds ratio and the respective 95% confidence interval. The independent variables with *p* < 0.20 in the univariable logistic regression model were selected for the multivariable model. The Hosmer–Lemeshow test was applied to evaluate the goodness of fit for logistic regression models. No imputation method was used for missing data.

Regarding the analyzes in which the time until death or recurrence are of interest, the survival analysis approach was used. The survival function was estimated using the Kaplan–Meier (1958) estimator and the estimated survival curves were compared using the log-rank test. In addition, a Cox proportional hazards regression model (Cox, 1972) was fitted to the data to describe the relationship between the independent variables with the time until death or recurrence. For the recurrence outcome, all variables with a *p* < 0.20 in the univariable Cox regression models were considered in the multivariable Cox regression model, while for the outcome of death, due the limitation of the number of events, we selected the variables chosen for the outcome of relapse, also considering their clinical relevance for this outcome. In all fitted models, the assumption of proportionality was assessed using Schoenfeld’s residuals (Schoenfeld, 1982; Grambsch and Therneau, 1994) and in all analyses we found evidence that the effect of covariates is constant over time, thus justifying the use of the Cox regression model.

The level of significance adopted was 5% in all analyses. Thus, results whose *p* < 0.05 are considered statistically significant. The software R version 3.6 was used in all analyses.

### 2.5. Assessment of Preoperative Clinical Response

The evaluation of the clinical response was determined by comparative analysis of the baseline with the physical, endoscopic examination and image at C1 with the preoperative (C2). Clinical response assessment was made within 8–10 weeks after the completion of chemoradiation. In addition, we evaluated the pathological response in comparison to the clinical staging established by baseline images. All parameters for radiological evaluation of the response (patient follow-up images) were obtained at the institution where the study was conducted and reviewed by a radiologist with experience in colorectal tumors.

### 2.6. Assessment of Pathological Response

The anatomopathological examination of the surgical material of patients submitted to TME was performed to assess the histological type, the tumor differentiation, the pathological stage of the TNM (ypTNM) and the degree of pathological regression of the tumor. The changes observed after NCRT were based on the tumor regression grading (TRG) system, divided into 4 categories according to the guidelines published by American Joint Committee on Cancer (AJCC) and the College of American Pathologists (CAP), with the adaptation of a 3-category classification scheme proposed by Ryan [32,33]. The TRG can be classified into 4 degrees: TRG 0 with no viable neoplastic cells, complete response; TRG 1 represents a moderate response, with rare neoplastic cells isolated or in small groups; TRG 2 is characterized by a minimal response with residual neoplasia with fibrosis; TRG 3 indicates absence of pathological response with extensive residual neoplasia, minimal necrosis. All material was subjected to review by a single pathologist with experience in gastrointestinal pathology, who was blinded to the analysis of CTCs image and analysis.

In addition, we evaluated the pathological response according to the TNM pathological staging (ypTNM) in comparison with the clinical staging established by baseline images to determine the downstaging obtained with the NCRT and, thus, patients were classified into: complete pathological response (ypT0N0), partial pathological (patients with downstaging, but still with residual disease) and without pathological response (without modification of pathological staging when compared to clinical staging).

### 2.7. Isolation, Quantification and Protein Expression Analysis of CTCs

To isolate and analyze CTCs, we used ISET^®^ (Isolation by SizE of Tumors cells, Rarecells, Paris, France). Briefly, EDTA tubes (8.0 mL BD Vacutainer, Franklin Lakes, NJ, USA) were used to collect peripheral blood samples from patients and were homogenized at room temperature for up to 6 h. Then, they were prepared as described previously (Troncarelli et al., 2019). After filtration, ISET membranes were stored at −20 °C until analysis.

To evaluate protein expression in CTCs, selected spots from the ISET membranes were subjected to a 24-well dual immunocytochemistry (ICC) assay (Polink DS-RR-Hu/Ms A Kit; GBI Labs, Bothell, WA, USA) with the following antibodies: anti-RAD23B (1:100 CSB-PA019260LA01HU; CusaBio, Wuhan, People’s Republic of China), anti-TYMS (1:230 WH0007298M1; Sigma-Aldrich, St. Louis, MO, USA), anti-CD45 (1:200 HPA000440; Sigma-Aldrich); anti-PD-L1 (1:300 Ab205921; Abcam, Cambridge, MA, USA), and anti-TGF-βRI (1:100 CSB-PA061850; Cusabio, Wuhan, People’s Republic of China. Briefly, the first step was to perform an antigen retrieval using an Antigen Retrieval Solution (Dako, Santa Clara, CA, USA). Following, cells were hydrated with 1 × Tris-buffered saline (TBS), 10 min and permeabilized with Triton X-100 (5 min). Rinsing was done with 1 × TBS and incubation to block endogenous peroxides with 3% hydrogen peroxide in the dark (15 min). Cells were incubated with primary antibodies diluted in 10% fetal calf serum in TBS (overnight). Primary antibody signals were amplified with rabbit horseradish peroxidase (HRP) polymer (GBI Labs) for 30 min and then, for 10 min with 3,3-diaminobenzidine (DAB). After amplification, an incubation with second antibody was made (2 h). Following, we added a rabbit AP polymer (GBI Labs) for 30 min, and GBI-Permanent Red (GBI Labs) for 10 min. Cells were staining with hematoxylin and examined in light microscopy (Research System Microscope BX61; Olympus, Waltham, MA, USA) (Figure 2). CTCs were counted per 1 mL blood, as previously described with statistical methods by Krebs et al. [34] and characterized according to 5 criteria: visible cytoplasm; nucleus size >18 µm, negativity for CD45 staining; hyperchromatic and irregular nuclei; a nuclear to cytoplasm ratio >80% [35]. It is important to take in consideration that we used at least 4 ISET spots to make our CTCs counts and that we collected at least 8 mL of blood of each patient. Therefore, the CTC counts are made per mL of blood, but more than 4mL were evaluated for each patient. CTCs were considered positive for the abovementioned antibodies expression if at least one cell was found staining in a spot. We emphasize that when CTC = 0, at any time, it was not possible to evaluate protein expression.

### 2.8. Survival Outcomes

In patients undergoing NCRT followed by surgery, the evaluation for adjuvant chemotherapy was performed by the attending physician. The chemotherapy regimen, if indicated, was defined by the team responsible for the patient. For these patients, the beginning of the follow-up was defined at the end of the adjuvant treatment.

In addition, for patients with no indication of adjuvant therapy or for patients who, after the end of NCRT, were not operated on due to evidence of a complete clinical response and, therefore, followed the “Watch and Wait” (WW) strategy, follow-up was initiated at this time.

The follow-up of all patients in the study consisted of clinical, laboratory and imaging assessments every three months.

Patients were followed up from the moment of the first CTC collection (C1) until the moment of death or last follow-up. Loss of follow-up was defined as the patient’s absence on two consecutive visits, with no outcome information after this period.

The cCR was defined as clinical disappearance (clinical examination with digital rectal examination), radiological (CT and/or pelvic MRI) and rectoscopy with the appearance of macroscopic disease associated with negative biopsy results in areas suspected or previously affected by the disease.

### 2.9. Prognosis Evaluation

All enrolled patients were followed until death or until the last follow-up date. Locoregional recurrence was defined as tumor recurrence of lymphatic vessels, anastomosis or adjacent organs. Distant metastasis was defined as the spread of the tumor out of the pelvic cavity. Disease-free survival (DFS) was calculated from the day of surgery (for patients undergoing tumor resection) or the day of evidence of cCR (biopsy and images; for patients who opted for organ preservation) until progression disease (distant metastasis or locoregional recurrence) or death. Overall survival (OS) was calculated from the date of the anatomopathological diagnosis until death or the date of the last follow-up.

## 3. Results

### 3.1. Demographic and Clinical Variables

From 2016 to 2020, a total of 166 blood samples were obtained from 63 patients with LARC undergoing treatment. A total of 70 patients were accrued and seven were excluded (three patients with problems in the CTC1 sample, three patients with loss of follow-up before surgery and one patient with modification of the therapeutic proposal). In addition, 7 patients out of the 63 who collected CTC1 (C1) and underwent NCRT did not undergo surgery (one patient with disease progression after NCRT, two patients refused surgery and four patients with a complete clinical response (cCR) being proposed the strategy of WW) (Figure 1).

The clinical features are reported in Table 1. Mean age was 56 years (range 34–92). Thirty-six patients were male (57.1%) and 27 female patients (42.9%). Most patients (80.9%; *n* = 51) had good performance status (ECOG 0) in the inclusion of the study. Moderately differentiated tumors (85.7%; *n* = 54 patients) were found in the diagnosis biopsy.

At the time of inclusion, 81% (*n* = 51) of the patients had clinical T3 and 60.3% (*n* = 38) clinical N1 according to the eighth edition of AJCC cancer staging manual. Carcinoembryonic antigen (CEA) levels >3.0 ng/mL were found in 52.5% (*n* = 31) and ≤3.0 ng/mL in 47.5% (*n* = 29) patients.

Regarding NCRT, all patients underwent the same radiotherapy protocol. Concerning chemotherapy, 54% (*n* = 34) of patients were submitted to infusional 5-FU and 46% (*n* = 29) to capecitabine.

Out of the 63 patients in the study, 4 had pCR response and of the 56 patients who underwent surgery after NCRT, 23.2% (*n* = 13) had pCR. Of the patients who underwent surgery in relation to the Degree of Tumor Regression (TRG), 23.2% were classified as TRG 0, 30.4 % as TRG 1, 31.1% as TRG 2 and 32.1% as TRG 3.

After surgery, 63.5% (*n* = 40) of the patients underwent adjuvant treatment with chemotherapy. In the follow-up of all patients in the study, 34.9% (*n* = 22) presented locoregional failure and distant metastasis. The main site of metastasis were lung 61.9% (*n* = 13) and liver 42.8% (*n* = 9). Palliative chemotherapy was given to 63.6% (*n* = 14) of patients and 50% (*n* = 11) underwent local treatment with surgery for metastasis resection.

Median follow-up was 32.33 months (95% confidence interval, 28.88–35.78). The mean for DFS was 32.15 months (95% confidence interval, 27.31–36.99) with 22 events and for OS, 46.83 months (95% confidence interval, 44.07–49.59) with 9 events. The median for DFS and OS was not reached.

### 3.2. CTCs Count

Of the 63 patients in the study, 166 samples were collected for isolation and quantification of CTCs; 63 samples at baseline (C1), 57 samples at first follow-up (C2) and 46 samples at second follow-up (C3) (Figure 1).

The detection rates of CTCs at different time points were: 88.9%, with a median of 2.0 (0–9.0 CTCs/mL) of blood at C1; 71.9% with a median of 0.6 (0–6.3 CTCs/mL) at C2 and at the time C3 was 54.3% with a median of 0.3 (0–4.6 CTCs/mL). 

### 3.3. Protein Expression in CTCs

The expression of RAD23B, TYMS, PD-L1 and TGF-βRI was evaluated in this order of priority according to the primary objective of the study at each time of collection (C1, C2 and C3) and the availability of cells (CTCs count > 0) in the sample and in the spots. The expression of RAD23B was evaluated in 56 samples in the first collection (of the 63 samples collected at C1, 7 samples had CTC = 0), in 41 samples at C2 (out of 57 samples 16 had CTC = 0) and 25 samples at C3 (from 46 collected, 21 had CTC = 0). At C1, 40% of patients (*n* = 22) had RAD23B positive and 60% (*n* = 34) negative. For C2 and C3, 39% (*n* = 16) and 0% (*n* = 0) of patients showed positive expression for RAD23B, respectively. TYMS expression was evaluated in a similar way to the expression of RAD23B in the same number of samples at each moment and was positive in 35% (*n* = 20) at C1, 39% (*n* = 16) at C2 and 0% (*n* = 0) at C3.

For TGF-βRI expression evaluation, 49 samples were available at C1 (of the 63 samples collected, 7 samples with CTC = 0 and 7 samples with no spot available), 17 samples at C2 (of the 57 samples collected, 16 with CTC = 0 and 24 without available spot) and only two samples at C3 (out of 46 samples, 21 with CTC = 0 and 23 without spot). TGF-βRI positive expression found in 20% (*n* = 10) of patients at C1, 18% (*n* = 3) at C2 and 0% (*n* = 0) at C3.

Finally, for PD-L1 analysis, 41 samples (out of 63 samples, 7 with CTC = 0 and 15 with no available spot) were made at C1, 39 samples at C2 (out of 57 samples, 16 samples with CTC = 0 and 2 samples without spot) and 25 samples at C3 (out of 46 samples, 21 samples with CTC = 0 and 17 samples with no spot for marking). PD-L1 expression was found in 19% (*n* = 8) of patients at C1, in no patient at C2 and in 16% (*n* = 4) at C3 (Figure 2).

### 3.4. Correlation between Clinical Characteristics, CTCs Kinetics and Protein Expression in CTCs with Pathological Response

#### 3.4.1. Clinical Characteristics

For patients with pCR or TRG 0 (23.2%; *n* = 13/56), 69.2% (*n* = 9) were female (*p* = 0.07), moderately differentiated adenocarcinomas (*p* = 0.09), CEA ≤ 3.0 (*p* = 0.11) and clinical T3 (*p* = 0.06); 61.5% (*n* = 8) of patients with pCR carried medium rectum tumors (*p* = 0.57) and underwent neoadjuvant treatment with 5-FU (*p* = 0.99). 

In the univariable logistic regression model, pCR was found more in females in relation to males (Odds Ratio (OR) 4.050; 95% confidence interval (CI), 1.064–15.409; *p* = 0.04), in cT3 in relation to cT2, (OR 0.083; 95% CI, 0.008–0.899; *p* = 0.04). However, in the multivariable logistic regression model, there was no significant difference for sex and clinical tumor staging (Table 2).

#### 3.4.2. CTCs Kinetics

For patients who had a CTC2 count (C2) higher than a CTC1 count (C1), only one patient (8.3%) had pCR and 6 (15.8%) showed no pCR (*p* = 0.99). Thus, when comparing patients with unfavorable kinetics (CTC2 > CTC1) with those with favorable kinetics (CTC2 ≤ CTC1), by univariable logistic regression model, there was a tendency of reduction in the chance of complete response (OR, 0.485; 95% CI, 0.052–4.487; *p* = 0.52), although without statistical significance (Table 2).

#### 3.4.3. RAD23B and TYMS Expression

ICC expression of RAD23B at C1 was detected in only one patient with pCR and in 54.1% (*n* = 20) of patients with no pCR. However, in 91.7% of patients with pCR (*p* = 0.014) RAD23B was absent. In the C2 collection, after the end of neoadjuvant with chemotherapy and radiotherapy, the absence of RAD23B expression correlated with the presence of pCR. In the second collection, of the 13 patients with pCR, 10 showed no RAD23B expression in the CTC (100%) and three patients had CTC = 0 (without the possibility to assess the RAD23B expression). On the other hand, for patients who did not obtain pCR with NCRT, 51.7% dissipation the expression of RAD23B in CTC at C2 (*p* = 0.06). Therefore, the absence of RAD23B expression at C1 and at C2 correlated directly with pCR, helping in the management of these patients (Figure 3). 

In the univariable (OR 0.077 (95% CI, 0.009–0.661); *p* = 0.019) and multivariable (OR 0.064 (95% CI, 0.006–0.751); *p* = 0.029) logistic regression models for pCR, we observed that RAD23B expression was associated with decreased chance of pCR as compared to no expression (Table 2).

In addition, 4 out of 63 patients had a complete clinical response and underwent an organ preservation strategy and, therefore, did not undergo surgery after the end of the NCRT. In these 4 patients, all did not show expression of RAD23B at C1 (*p* ≤ 0.001) and at C2, three patients with the presence of CTC had no expression of RAD23B (*p* ≤ 0.001) reinforcing the correlation of the response to NCRT with the absence of RAD23B. One patient at C2 did not have isolated CTC, making it impossible to assess RAD23B expression. 

We evaluated the expression of TYMS by ICC in CTCs at C1. We evidenced that among patients with pCR, 33.3% (*n* = 4) had positive expression of TYMS. For patients who did not obtain a pCR, 37.8% (*n* = 14) had positive expression (*p* = 0.99). In the univariable logistic regression model, the presence of TYMS expression showed no correlation (OR 0.821; 95% CI, 0.208–3.239; *p* = 0.77) with the absence of pCR (Table 2). On the other hand, when we evaluated the expression of TYMS after NCRT (C2), 90% of patients with pCR did not have TYMS expression (*p* = 0.057). By univariable logistic regression model, the presence of TYMS expression in the CTC presented OR of 0.119 (95% CI, 0.013–1.064; *p* = 0.057) for pCR. 

In the combined analysis of RAD23B and TYMS expression in CTCs at C1, we showed that among patients with pCR, 66.7% (*n* = 8) had no RAD23B/ TYMS expression; for patients with no pCR, 29.7% (*n* = 11) had RAD23B/ TYMS positive and 24.3 % (*n* = 9) had negative TYMS and positive RAD 23 B (*p* = 0.029). In the second collection (C2), for patients with pCR, 90% (*n* = 9) had no RAD23B/TYMS staining and for patients with no pCR, 41.4% (*n* = 12) had RAD23B/TYMS staining (*p* = 0.029). These data reinforce the correlation between the absence of RAD23B/TYMS expression in CTCs and the response before and after NCRT.

### 3.5. Correlation between CTCs Kinetics and Protein Expression in CTCs with DFS and OS

Patients who presented unfavorable CTC kinetics during neoadjuvant treatment (CTC2 > CTC1) had worse DFS (median 9.11 months; 95% CI 6.21–12.00) compared with those with favorable kinetics (CTC2 ≤ CTC1) (median 33.85 months; 95% CI 28.63–30.07) (*p* = 0.00025). The unfavorable kinetics also correlated with poor OS. For patients with CTC2 > CTC1, the median OS was 35.57 months (95% CI 26.10–45.05) and for patients with favorable kinetics the mean OS was 47.74 months (95% CI 44.93–50.56) (*p* = 0.0036) (Figure 4). 

In the univariable analysis for patients with CTC2 > CTC1, we demonstrated worse DFS and OS, respectively (hazard ratio (HR) 7.042; 95% CI, 2.111–23.491; *p* = 0.001) and (HR 4.954; 95% CI, 1.141–21.504; *p* = 0.033). Cox’s multivariable regression model suggests that CTC2 > CTC1 remained an independent predictor of DFS and OS, respectively (HR 12.463; 95% CI, 2.998–51.811; *p* = 0.001) and (HR 5.503; 95% CI, 1.255–24.122; *p* = 0.024) (Table 3 and Table 4).

### 3.6. Protein Expression in CTCs x DFS and OS

We evaluated the expression of RAD23B, TYMS, PD-L1 and TGF-βRI at each time of collection according to the availability of cells and/or spots of ISET membranes for the immunocytochemistry.

The positive expression of RAD23B at C1 and C2 collection, although shown to be correlated with the absence of a pCR, had no significant impact on the prognosis. At C1, patients with positive expression for RAD23B had a mean DFS of 30.20 months versus 34.87 months for patients without expression of RAD23B (*p* = 0.28) and at time C2 it was 25.83 months for patients with positive expression versus 32.64 months for negative (*p* = 018). For OS at C1, patients with positive RAD23B showed 46.79 months versus 47.98 months with negative expression (*p* = 0.52) and at C2 it was 43.65 months for patients with positive expression versus 47.49 for patients with negative RAD23B (*p* = 0.23). For univariable analysis of the expression of RAD23B in the first collection, DFS (HR 1.655; 95% CI, 0.651–4.206; *p* = 0.29) and OS (HR 1.602; 95% CI, 0.375–6.847; *p* = 0.52) showed a tendency towards a worse prognosis. The expression of TYMS showed no difference in DFS and OS.

A combined analysis of the three collections (C1, C2 and C3) was made for the expression of PD-L1 and TGF-βRI, that is, we analyzed the positivity for these proteins at any time. From the total of 166 samples from the three moments of the study, we were able to evaluate the combined expression for PD-L1 and TGF-βRI, respectively, in 63% (*n* = 105) and 41% (*n* = 68) samples. 

The combined expression of PD-L1 in CTCs showed a trend to better DFS of 32.91 months for patients with positive expression for PD-L1 and 30.77 months for patients with no expression (*p* = 0.32). For OS, patients with negative expression for PD-L1 had OS of 46.40 months versus 45.07 months for those with positive expression (*p* = 0.64). 

When we evaluated the combined expression of TGF-βRI, we noticed a tendency that its expression in the CTC at any time of the treatment (C1 or C2 or C3) could be correlated with a worse DFS, that is, patients who presented positivity to the expression of TGF-βRI had an average DFS of 22.36 months versus 34.47 months for patients with negative expression (*p* = 0.059) (Figure 5). For OS, patients with positive expression had an average of 44.34 months versus 47.27 months for patients without expression of TGF-βRI on CTC (*p* = 0.25). 

## 4. Discussion

In the present study, we reinforce the previously published data, which favors the applicability of CTC in the management of the LARC patients. Unfortunately, the response to NCRT is heterogeneous, with only about 20% of patients with LARC undergoing NCRT presenting a pCR and 40% of patients with no response or minimal response [36,37,38,39]. In our study, we found this heterogeneity, with 13 patients (23.2%) with pCR and 44.6% (*n* = 25) with poor response (TRG 2 and 3) to NCRT. 

Important studies have demonstrated the relevance of pCR, as an outcome that must be achieved in NCRT, as it is associated with good DFS and 5-year OS [11,40]. For patients with pCR, especially for tumors of the distal rectum, the WW strategy is quite attractive and safe [10,41]. However, the clinical response presents poor correlation with the pathological response [42,43,44]. Therefore, in addition to endoscopy and imaging, new tools are need to more accurately predict response. In view of the preservation strategy, there is a rationale for treatment intensification with the aim of increasing the response to treatment. The OPRA study [45], whose preliminary results were presented at ASCO 2020, demonstrated that TNT is important for WW strategy. 

Ionizing Radiation (IR) is an important pillar of NCRT for patients with LARC, with the aim of providing cell damage and death [23,46]. However, not all tumor cells have radiosensitivity, since there are mechanisms capable of effectively repairing their damaged DNA [47,48]. Therefore, understanding these mechanisms is fundamental to search for possible biomarkers of resistance to this therapeutic strategy. Thus, as we have already described, the expression of RAD23B could be studied as a possible biomarker for resistance to IR [18]. Here, we demonstrated RAD23B expression was associated with a lower chance of pCR compared to no expression in the univariable analysis (OR 0.077; 95% CI; 0.009–0.661; *p* = 0.019) and multivariable logistic regression (OR 0.064 (95% CI), 0.006–0.751; *p* = 0.029). Despite the wide confidence interval, mainly justified by the number of patients involved in the regression model (*n* = 47), this data demonstrates that the presence of RAD23B in CTC1 implies a 93.6% chance of not presenting pCR to NCRT. In other words, in our cohort, the absence of RAD23B expression at the baseline and/or after NCRT reinforces the correlation with the pCR. In the 13 patients with pCR and in the four patients with cCR, the absence of RAD23B expression correlated with response to treatment. As far as we know, our results are the first to show the possible role of this molecule in rectum cancer and its analysis in CTCs can be an additional tool to complement the response assessment exams already adopted in medical practice, reinforcing the evidence of cCR for WW strategy.

On the other hand, for non-responders, the standard NCRT must be reevaluated in order to avoid unnecessary toxicities or, eventually, establish new treatment strategies capable of optimizing the response of these patients, such as the adoption of more intense treatments according to the reasoning proposed by the TNT’s strategy [7,49]. In view of this, numerous efforts with studies evaluating the intensification of NCRT are underway and the recently published RAPIDO study and the PRODIGE 23 phase III study reinforced the importance of the TNT strategy with the addition of chemotherapy (FOLFOX/CAPOX or FOLFIRINOX) in increasing the rate of pCR and reducing about 7% of distant metastases [8,9]. However, we reinforce with our results that the risk stratification must be improved and the incorporation of the liquid biopsy in these studies must be considered, in order to avoid overtreatment for some patients. In addition, the identification of markers such as RAD23B and TYMS in CTCs could identify patients who are radioresistant to standard NCRT treatment. With this information, we could, in future clinical trials, assess whether in these patients we could omit radiotherapy. This could minimize possible adverse effects provided by NCRT and thereby surrogate for more intense chemotherapy regimens for patients whose CTCs express RAD23B.

Another relevant point of our study was that we demonstrated the importance of analyzing the kinetics of CTCs during NCRT. Previous studies of our group had already demonstrated this relevance for patients with metastatic colorectal cancer [50,51], that is, the quantification of CTC at different times helps us to stratify patients with unfavorable kinetics (CTC2 > CTC1) versus favorable kinetics (CTC2 ≤ CTC1). However, for localized disease, we still have few studies. Despite failing to demonstrate a difference with the pCR, patients with favorable kinetics tend to evolve with a greater chance of pCR compared to those with unfavorable one. Therefore, CTC kinetics can be applied in larger studies to corroborate the prediction of response. Considering DFS and OS, patients who presented unfavorable kinetics (CTC2 > CTC1), that is, there was an increase in the quantification of CTC during treatment with NCRT, presented inferior DFS and OS in relation to those with favorable kinetics (CTC2 ≤ CTC1). Therefore, in view of these results, we could discuss to intensify treatment with chemotherapy for these patients. For the analysis of TGF-βRI and PD-L1 we had difficulties due to the limited number of available spots from ISET membranes to evaluate these markers in patients CTCs. Even with these limitations, we were able to demonstrate that the expression of TGF-βRI during treatment may, despite the lack of statistical significance, be related to worse prognosis. This result corroborates studies that describe that higher levels of TGF-βRI may be indicative of distant metastases with worse outcomes. Maybe our work may pave the way for future trials to better evaluate TGF-βRI in rectal cancer. Concerning PD-L1, the clinical benefit of the blocking this molecule in some tumors, such as rectal cancer, still remains uncertain. Some previous studies demonstrate that the expression of PD-L1 in the tumor during neoadjuvant treatment can be modified and have a prognostic impact [27,28,29]. Here, unfortunately due, probably, the limitation of the samples, we could not demonstrate any prognostic difference. However, we observed that the positive expression of PD-L1 in CTCs occurs dynamically with treatment, as of 105 samples evaluated, 8, 0 and 4 were positive for PD-L1 at C1, C2 and C3, respectively. Therefore, by demonstrating in our study the possibility of dynamically analysis of this expression in CTCs, we could evaluate new therapeutic perspectives for these patients with the use of immunotherapy guided by the expression of PD-L1 in in these cells, suggesting a NCRT strategy combined with blockade of the PD-1/PD-L1 pathway. At the American Society of Clinical Oncology (ASCO) congress, that happened in June 2021, Salvatori et al. reported the addition of Avelumab (anti-PD-L1) in patients with LARC undergoing NCRT. Although they did not search for PDL-1 expression in tumor sample, the authors demonstrated that the combination of NCRT with Avelumab showed promising activity and a viable safety profile that should be explored in future clinical studies [52].

We are aware of the limitations of our study, mostly, the small sample size, but we believe that despite of this, our results must be explored in future clinical trials, as we showed that the evaluation of the expression of RAD23B and TYMS associated with the evaluation of CTC kinetics may facilitate the personalized treatment of rectal cancer. In Table 5 we describe the main points that strengthen our results and our limitations. Despite the important impact on local control provided by NCRT, remote recurrence is still a problem and around 20–30% of patients have distant metastases after curative treatment [6,53], so, novel tools like liquid biopsy are very promising.

## Figures and Tables

**Figure 1 cells-10-01539-f001:**
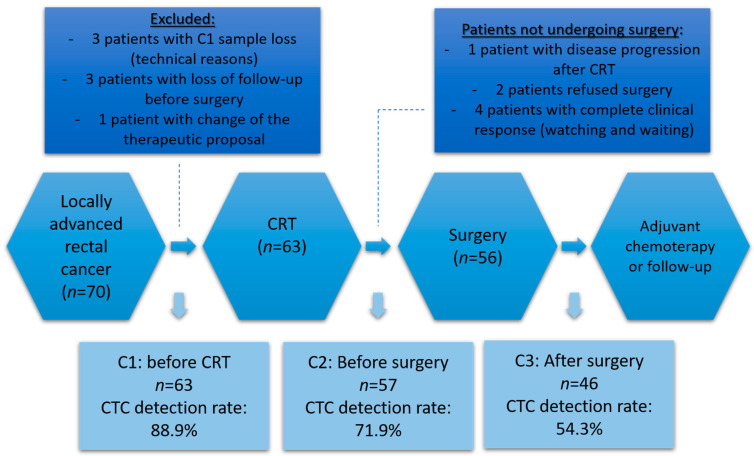
Flowchart showing the timing of each CTC collection (C1, C2 and C3) and its rate of detection during the study and the reasons for excluding patients (at C1) and not undergoing surgical approach (at C2). Abbreviations: CRT, chemoradiotherapy; CTC, circulating tumor cell.

**Figure 2 cells-10-01539-f002:**
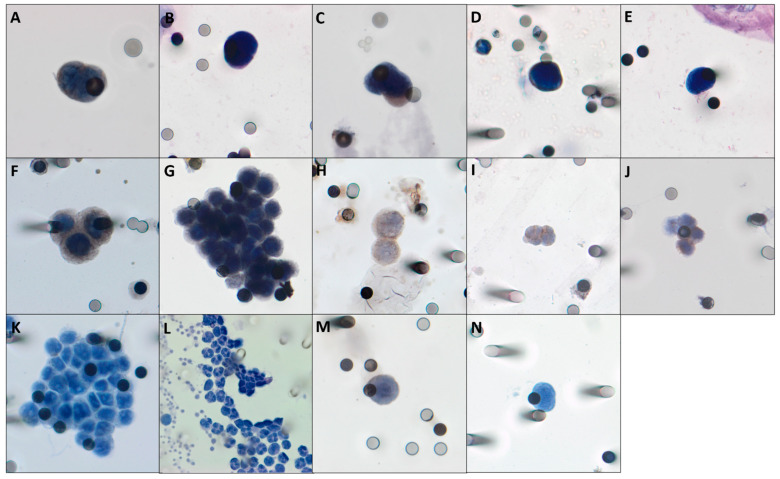
Photomicrographs of CTCs isolated from LARC patients, stained with anti-PD-L1 (**A**), anti-RAD-23B (**B**), anti-TGF-βRI (**C**) and anti-TYMS (**D**). In (**E**), a CTC with hematoxylin, without any antibody staining. Following, in boxes (**F**–**H**), positive controls (cell lineages, that accordingly to Protein Atlas (https://www.proteinatlas.org/ accessed on 13 December 2016) express the studied antibodies, spiked in healthy individual blood). In (**F**), FADU lineage, expressing PD-L1. In box (**G**), HCT-8 cells expressing RAD23B. In (**H**), A549 cells with TGF-βRI staining; in (**I**,**J**), leucocytes staining for TYMS and CD45, respectively. In boxes (**K**,**M**,**N**), negative controls (cell lineages, that accordingly to Protein Atlas (https://www.proteinatlas.org/ accessed on 13 December 2016) do not express or express very low levels of the studied antibodies, spiked in healthy individual blood). In (**K**), PC3 lineage without any expression of PD-L1. In (**M**), MCF7 without TGF-βRI staining and in (**N**), U87MG without TYMS expression. In (**L**), leucocytes from healthy individuals without RAD23B staining. For negative control of CD45, we used A549 lineage.

**Figure 3 cells-10-01539-f003:**
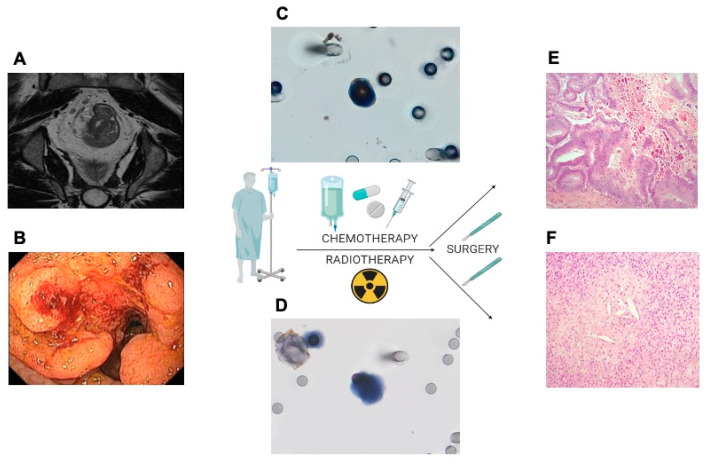
Illustration of a patient with LARC (**A**,**B**) evidenced in baseline imaging exams with RAD23B positive in CTC (**C**) with no response TRG 3 (**E**) and RAD23B negative on CTC (**D**) with pCR TRG 0 (**F**). (**A**): Rectal lesion on pelvic MRI with extension to perirectal fat with compromised mesorectal lymph nodes; (**B**): Rectoscopy with evidence of ulcerative injury to the rectum; (**C**): CTC collected at the time C1 or C2 with positive RAD23B; (**D**): CTC collected at the time C1 or C2 with negative RAD23B; (**E**): Anatomopathological examination of the surgical material indicates absence of pathological response with extensive residual neoplasia (TRG 3); (**F**): Anatomopathological examination of the surgical material with evidence of pCR (TRG 0). Abbreviations: LARC, locally advanced rectal carcinoma; RAD23B, RAD23 homolog B; CTC, circulating tumor cell; TRG, tumor regression grading; MRI, magnetic image resonance; C1, collection 1; C2, collection 2.

**Figure 4 cells-10-01539-f004:**
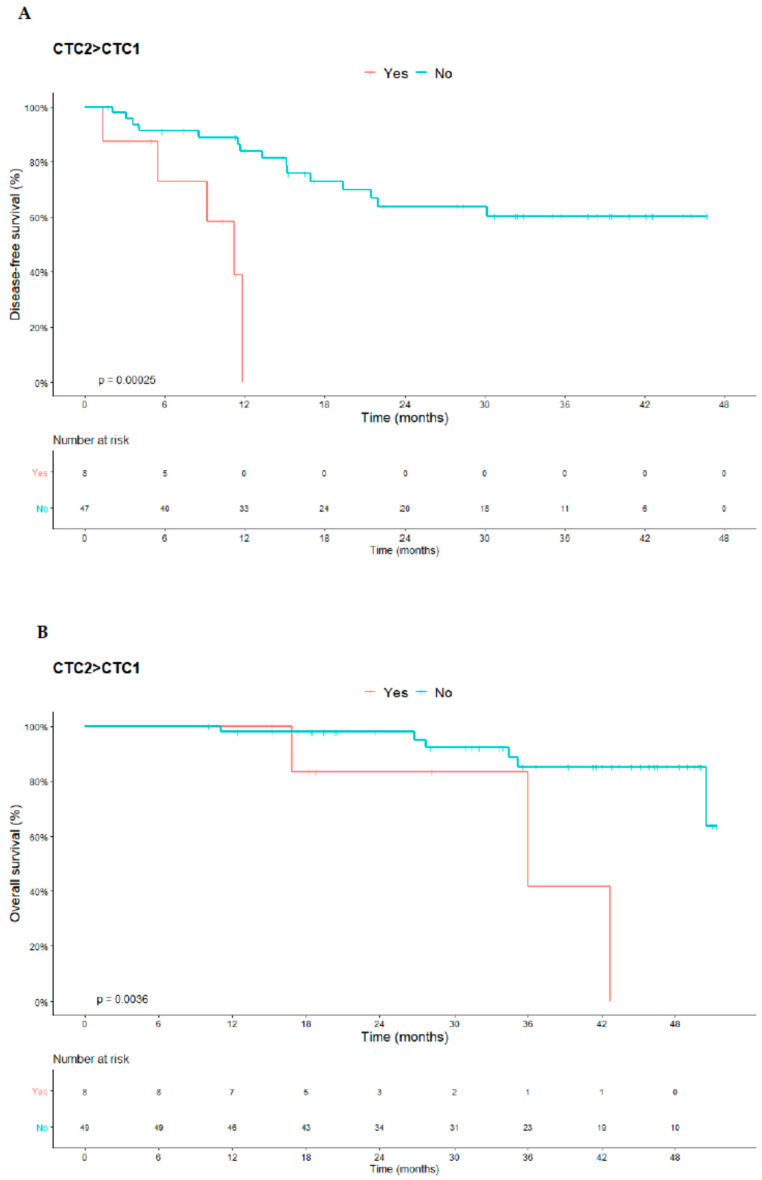
Analysis of DFS and OS of patients with LARC in relation to the kinetics of CTCs (CTC2 > CTC1). (**A**): DFS median of 9.11 months of patients with unfavorable kinetics (CTC2 > CTC1) versus favorable kinetics (CTC2 < CTC1) with a median of 33.85 months; (**B**): OS of patients with unfavorable kinetics (CTC2 > CTC1) with a median of 35.57 months versus favorable kinetics (CTC2 ≤CTC1) with a median of 47.74 months. Abbreviations: CTCs, circulating tumor cells; DFS, disease-free survival; OS, overall survival; LARC, locally advanced rectal carcinoma.

**Figure 5 cells-10-01539-f005:**
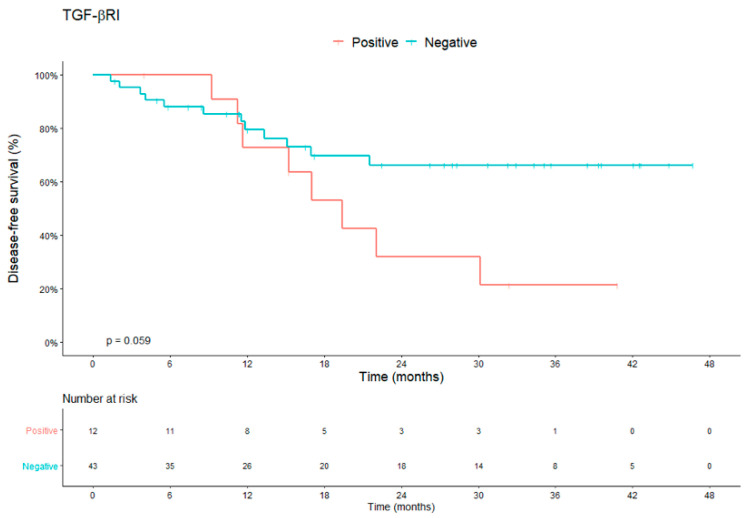
DFS analysis of patients with LARC for combined TGF-βRI expression in CTC during treatment (C1, C2 or C3). Median DFS of 22.36 months for patients with positive expression for TGF-B and 34.47 months for patients with negative expression (*p* = 0.059). Abbreviations: CTCs, circulating tumor cells; DFS, disease-free survival; LARC, locally advanced rectal carcinoma.

**Table 1 cells-10-01539-t001:** Clinical-pathological aspects, epidemiology, and treatment of 63 patients with locally advanced rectal cancer undergoing neoadjuvant treatment.

Clinical Features	N	%
Total number of patients	63	100
Age (years) at recruitment		
Median	56	
Range (min–max)	34–92	
Gender		
Male	36	57.1
Female	27	42.9
ECOG Performance Status		
0	51	80.9
1	12	19.1
Tumor differentiation		
Well differentiated	7	11.1
Moderately differentiated	54	85.7
Poorly differentiated	2	3.2
Distance of tumor from dentate line (cm)		
<4	34	54
4–8	29	46
Median	4	
Range (min-max)	0.5–8	
Clinical Tumor Stage (cT)		
cT1 or cT2	6	9.5
cT3	51	81
cT4	6	9.5
Clinical Nodal Stage (cN)		
cN0	5	7.9
cN1	38	60.3
cN2	20	31.8
CEA in diagnosis (ng/mL; *n* = 61)		
>3	32	52.5
≤3	29	47.5
Ca 19.9 in diagnosis (U/mL; *n* = 55)		
>37	10	18.2
≤37	45	81.8
Neoadjuvant Chemotherapy		
Capecitabine	29	46
5-Fluorouracil (5-FU)	34	54
Pathological Tumor Stage (pT; *n* = 56)		
ypT0-ypT2	34	60.7
ypT3-ypT4	22	39.3
Pathological Nodal Stage (pN; *n* = 56)		
ypN0	45	80.4
ypN1-ypN2	11	19.6
Degree of tumor regression (TRG) (*n* = 56)		
0	13	23.2
1	17	30.4
2	18	32.1
3	7	12.5
Unknown	1	1.8
Pathological Complete Response (*n* = 56)		
Yes	13	23.2
No	43	76.8
MMR (*mismatchrepair*)		
Proficient	38	60.3
Deficient	0	0
Unknown	25	39.7
Adjuvant Chemotherapy		
Yes	40	63.5
No	23	36.5
Locoregional failure and distant metastasis		
Yes	22	34.9
No	41	65.1

Abbreviations: CEA, carcinoembryonic antigen; ECOG: Eastern Cooperative Oncology Group.

**Table 2 cells-10-01539-t002:** Univariable and multivariable logistic regression models for complete pathological response.

	Category	Univariable Logistic Regression Model			Multivariable Logistic Regression Model		
		OR	95% CI	*p*	OR	95% CI	*p*
Gender	Male	1.0			1.0		
	Female	4.050	1.064–15.409	0.040	3.180	0.626–16.163	0.163
CEA (ng/mL)	≤3.0	1.0			1.0		
	>3.0	0.329	0.087–1.247	0.102	0.346	0.065–1.858	0.216
Clinical Tumor Stage	cT2	1.0			1.0		
	cT3	0.083	0.008–0.899	0.041	0.568	0.032–10.081	0.700
	cT4	0.067	0.003–1.509	0.089	0.936	0.020–42.968	0.973
Chemotherapy	5-FU	1.0					
	Capecitabine	0.919	0.257–3.292	0.897	---	---	---
RAD23B CTC1	Negative	1.0			1.0		
	Positive	0.077	0.009–0.661	0.019	0.064	0.006–0.751	0.029
TYMS CTC1	Negative	1.0					
	Positive	0.821	0.208–3.239	0.779	---	---	---
CTC2 > CTC1	No	1.0	0.052–4.487	0.524	---	---	---
	Yes	0.485					

Abbreviations: CTC, circulating tumor cells; CI (95%), confidence interval; OR, odds ratio; CEA, carcinoembryonic antigen; cT, clinical tumor stage; 5-FU, 5-fluorouracil.

**Table 3 cells-10-01539-t003:** Estimate of the parameters of the univariable and multivariable Cox regression models for disease-free survival of patients with LARC undergoing NCRT.

	Category	Univariable Cox Regression Model			Multivariable Cox Regression Model		
		HR	95% CI	*p*	HR	95% CI	*p*
pCR	No	1.0			1.0		
	Yes	0.138	0.018–1.029	0.053	0.388	0.146–1.032	0.058
CEA (ng/mL)	≤3.0	1.0			---		
	>3.0	2.527	0.969–6.590	0.058	---	---	---
CTC2 > CTC1	No	1.0			1.0		
	Yes	7.042	2.111–23.491	0.001	12.463	2.998–51.811	0.001
RAD23B CTC1	Negative	1.0			---		
	Positive	1.655	0.651–4.206	0.290	---	---	---
TGF-βRI combined CTC	Negative	1.0			1.0		
	Positive	2.314	0.944–5.676	0.067	2.235	0.830–6.018	0.111
Adjuvant Chemotherapy	No	1.0			---	---	---
	Yes	0.537	0.217–1.348	0.185	---	---	---

Abbreviations: CI (95%), confidence interval; HR, hazard ratio; CTC, circulating tumor cells; CEA, carcinoembryonic antigen; pCR, complete pathological response; TGF-βRI combined CTC, Expression of TGF-βRI in CTC1 and/or CTC2 and/or CTC3.

**Table 4 cells-10-01539-t004:** Estimate of the parameters of the univariable and multivariable Cox regression models for overall survival of patients with LARC undergoing NCRT.

	Category	Univariable Cox Regression Model			Multivariable Cox Regression Model		
		HR	95% CI	*p*	HR	95% CI	*p*
pCR	No	1.0			1.0		
	Yes	0.440	0.054–3.579	0.443	0.378	0.081–1.771	0.217
CEA (ng/mL)	≤3.0	1.0			---		
	>3.0	2.576	0.498–13.333	0.259	---	---	---
CTC2 > CTC1	No	1.0			1.0		
	Yes	6.502	1.531–27.607	0.011	5.503	1.255–24.122	0.024
RAD CTC1	Negative	1.0			---		
	Positive	1.602	0.375–6.847	0.525	---	---	---
Adjuvant Chemotherapy	No	1.0			---		
	Yes	0.379	0.097–1.476	0.162	---	---	---
TGF-βRI combined CTC	Negative	1.0			1.0		
	Positive	2.117	0.567–7.908	0.265	1.849	0.397–8.611	0.433

Abbreviations: CI (95%), confidence interval; HR, hazard ratio; CTC, circulating tumor cell; CEA, carcinoembryonic antigen; pCR, complete pathological response; TGF-βRI combined CTC, Expression of TGFβ-RI in CTC1 and/or CTC2 and/or CTC3.

**Table 5 cells-10-01539-t005:** Description of the main points that strengthen the study and its limitations.

Strengths of the Future Perspectives	Limitations
Molecular and dynamic evaluation of proteins in circulating tumor cells allow complementary predictive and prognostic evaluation;	The proteins TGF-βRI and PD-L1 were not evaluated in all CTC samples;
CTC kinetics (favorable kinetics versus unfavorable kinetics) enable prognostic assessment during treatment and thus evaluate more intensive treatment regimens;	Probably because of the limitation of the number of samples, we could not demonstrate any prognostic difference in relation to PD-L1 expression.
The absence of RAD23B expression in CTCs may identify radiosensitive patients and improve selection for sphincter preservation strategy;	
The expression of TGF-βRI in CTCs may assist the prognostic evaluation of locally advanced rectal cancer (LARC) patients and be used as target molecule for future clinical trials;	Sample size, volume of blood collected and availability of ISET spots should be explored in future clinical trials for the incorporation of liquid biopsy in the therapeutic management of patients with rectal cancer undergoing neoadjuvant treatment.
Serial blood collection during treatment makes the dynamic analysis of the expression of PD-L1 in CTCs feasible and, therefore, permit to evaluate the possible incorporation of immunotherapy in the neoadjuvant treatment of LARC.

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
