# Peer review of "Molecular and Dynamic Evaluation of Proteins Related to Resistance to Neoadjuvant Treatment with Chemoradiotherapy in Circulating Tumor Cells of Patients with Locally Advanced Rectal Cancer"

_cells, 2021, doi:10.3390/cells10061539_

Round 1

Reviewer 1 Report

The manuscript presented by Dr. Virgílio Souza e Silva and colleagues is performed enrolling patients affected by locally advanced rectal
cancer (LARC).  
For the study, three serial blood samples were collected. This information is well added also in a figure. Probably, the authors should improve only the resolution of figure 1 that appears grainy. 
The goals are: To determine whether the absence of RAD23B and TYMS expression in CTC would correlate with pCR to NCRT and identify possible responders to treatment; to assess the CTC kinetics between baseline (C1) and after NCRT (C2) in addition to the evaluation of the immune response markers expression such as TGF-βRI and PD-L1 with recurrence and survival rate. At this point, the authors should introduce better PD-L1 also citing its important role in immune-checkpoint since now these data are very appealing. 

In the end, the authors admit that there are some limits to their research. They should add a table introducing the strengths of the future perspectives and the limitations of this approach based on the present literature and their data. What's the point of strength of their research? They should better add this point.

Author Response

Reviewer 1

Comment 1: The manuscript presented by Dr. Virgílio Souza e Silva and colleagues is performed enrolling patients affected by locally advanced rectal
cancer (LARC).  
For the study, three serial blood samples were collected. This information is well added also in a figure. Probably, the authors should improve only the resolution of figure 1 that appears grainy. 

Answer to comment 1: We thank this referee for the careful evaluation of version R1 of our manuscript and for their useful suggestions. We improved this figure.

Comment 2:

The goals are: To determine whether the absence of RAD23B and TYMS expression in CTC would correlate with pCR to NCRT and identify possible responders to treatment; to assess the CTC kinetics between baseline (C1) and after NCRT (C2) in addition to the evaluation of the immune response markers expression such as TGF-βRI and PD-L1 with recurrence and survival rate. At this point, the authors should introduce better PD-L1 also citing its important role in immune-checkpoint since now these data are very appealing. 

Answer to comment 2: thank you for your suggestion, it improved paper. We added a sentence about PD-L1 in the discussion section.

Comment 3: In the end, the authors admit that there are some limits to their research. They should add a table introducing the strengths of the future perspectives and the limitations of this approach based on the present literature and their data. What's the point of strength of their research? They should better add this point.

Answer to comment 3: thank you for this suggestion. We made a table and added it to the paper (Table 5). We hope you appreciate it.

Reviewer 2 Report

The study is well structured with the limit of the  small sample size so we cannot draw definitive conclusion. The paper is  well written and represents the consequential update of the previous work. I suggest to check some english  expressions (eg: degree of tumor should be more appropriate define as tumor grading or differentiation).

In the discussion I'd explain how RADB23 and TYMS expression could be affected by intensification strategy as induction/consolidation chemotherapy+CTRT in the total neoadjuvant treatment (cited in the background and at the begining of discussion).

Author Response

Reviewer 2

Comment 1: The study is well structured with the limit of the small sample size so we cannot draw definitive conclusion. The paper is  well written and represents the consequential update of the previous work. I suggest to check some english  expressions (eg: degree of tumor should be more appropriate define as tumor grading or differentiation).

Answer to comment 1: Thank you for your positive comments about our study. It stimulates us to continue our work. We made an English review and corrected some expressions.

Comment 2: In the discussion I'd explain how RADB23 and TYMS expression could be affected by intensification strategy as induction/consolidation chemotherapy+CTRT in the total neoadjuvant treatment (cited in the background and at the begining of discussion).

Answer to comment 2: after your comment, we improved the discussion about the role of these molecules in selecting patients. We hope you appreciate it.
